# A Survey on Fluorinated Greenhouse Gases in Taiwan: Emission Trends, Regulatory Strategies, and Abatement Technologies

**Wen-Tien Tsai [1],* and Chi-Hung Tsai [2]**

[1] Graduate Institute of Bioresources, National Pingtung University of Science and Technology, Pingtung 912, Taiwan
[2] Department of Resources Engineering, National Cheng Kung University, Tainan 701, Taiwan; ap29fp@gmail.com
* Correspondence: wttsai@mail.npust.edu.tw; Tel.: +886-8-7703202

**Abstract:** Fluorinated greenhouse gases (F-gases), including hydrofluorocarbons (HFCs), perfluorocarbons (PFCs), sulfur hexafluoride ($SF_6$), and nitrogen trifluoride ($NF_3$), are used in a variety of applications, but they are potent greenhouse gases (GHGs). Therefore, they have been blanketed into the list of items to be phased out under international protocols or treaties. During the desk research, the updated statistics of Taiwan's National Inventory Report (NIR) were used to analyze the trends of F-gases (i.e., HFCs, PFCs, $SF_6$, and $NF_3$) emissions during the period of 2000–2020. Furthermore, the regulatory strategies and measures for the reduction of the four F-gas emissions will be summarized to be in accordance with the national and international regulations. With the rapid development in the electronics industry, the total F-gas emissions indicate a significant increase from 2462 kilotons of carbon dioxide equivalents ($CO_{2eq}$) in 2000 to the peak value (i.e., 12,643 kilotons) of $CO_{2eq}$ in 2004. However, it sharply decreased from 10,284 kilotons of $CO_{2eq}$ in 2005 to 3906 kilotons of $CO_{2eq}$ in 2020 due to the ongoing efforts of the regulatory requirements and the industry's voluntary reduction in time sequence. It was also found that the most commonly used method for controlling the emissions of F-gases from the semiconductor and optoelectronic industries in Taiwan is based on the thermal destruction-local scrubbing method.

**Keywords:** fluorinated greenhouse gas; emission trend analysis; regulatory policy; abatement technology

## 1. Introduction

Certain gases are called greenhouse gases (GHGs) due to their abilities to enhance infrared (IR) radiation in the lower (tropospheric) atmosphere, thus leading to the heating of the surface of the earth, or the so-called greenhouse effect. Naturally, the major GHGs include water vapor ($H_2O$), carbon dioxide ($CO_2$), methane ($CH_4$), nitrous oxide ($N_2O$), and ozone ($O_3$) [1]. Although chlorofluorocarbons (CFCs) and hydrochlorofluorocarbons (HCFCs) were phased out under the Montreal Protocol on Substances that Deplete the Ozone Layer [2], the significant emissions of these synthetically chemical alternatives, including hydrofluorocarbons (HFCs), perfluorocarbons (PFCs), sulfur hexafluoride ($SF_6$), and nitrogen trifluoride ($NF_3$), have been observed since the 1990s to be consistent with the enhancement of global warming or greenhouse effect due to their radioactively active features with the high global warming potential (GWP) [3–6]. More importantly, the temperature rise of 1.5 °C on earth could negatively affect the climate system and ecosystem, causing extreme weather events, shifting wildlife populations and habitats, rising sea levels, and increased disease/epidemic risks [7].

To negotiate the voluntary reduction of GHG emissions from anthropogenic activities, the Third Conference of the Parties (COP-3) to the United Nations Framework Convention on Climate Change (UNFCCC) was held from 1–11 December 1997 in Kyoto (Japan). Based

on their high GWP values (i.e., hundreds to tens of thousands of times stronger than $CO_2$), HFCs, PFCs, and $SF_6$ were included in the basket of the major GHGs for negotiation. Furthermore, $NF_3$ was also mandated to be included in the National Inventory Report (NIR) in the eighteenth session (COP18) of the UNFCCC (held in Dec. 2012) [8]. These fluorinated GHGs (F-gases) are man-made gases that are mostly used as industrial and commercial products like refrigerant, etchant, blowing agent, cleaning solvent, and coolant fluid [8–11]. To further reduce the emissions of F-gases, the Kigali Amendment, which was signed on 15 October 2016 and entered into force on 1 January 2019 [12,13], added HFCs to the list of controlled substances under the Montreal Protocol. For the developed countries, they are committed to reducing the use of HFCs by 45% by 2024 and by 85% by 2036, compared to their use between 2011 and 2013. On the other hand, the European Union (EU) has promulgated the new F-gases regulation, which has been in effect since 1 January 2015, replacing its original regulation adopted in 2006 [8,10,14]. The current regulation focuses on limiting the total amounts of the most commonly used F-gases (i.e., HFCs) sold, using low-GWP alternatives, and banning the use of F-gases in many new types of equipment.

In response to international regulations and protocols, the Taiwan government has taken initiatives to prepare the national GHG inventory reports since 1998 in accordance with the guidelines of the Intergovernmental Panel on Climate Change (IPCC). Currently, the statistical data on national GHG emissions have been established by ranging the period from 1990 to 2020 [15]. Table 1 summarizes the atmospheric lifetime, radiative efficiency, and global warming potential (GWP) of fluorinated greenhouse gases (F-gases) [3], which are mainly used in a variety of Taiwan's industries [16]. Herein, radiative efficiency is a measure of greenhouse strength based on the change in radiative forcing per change in atmospheric concentration of a gas (Watts per meter square per part per billion, $Wm^{-2}ppb^{-1}$). The index GWP is defined as a comparative measure of how much energy the emissions of 1 ton of a gas will absorb over a given period of time (e.g., 100 years) relative to the emissions of 1 ton of carbon dioxide ($CO_2$). According to the data on atmospheric lifetime, most F-gases are long-lived GHGs, especially in PFCs, $SF_6$, $NF_3$, and few HFCs like HFC-23. On the other hand, the regulatory establishment may be the most important and efficient tool for mitigating GHG emissions. In this regard, the central competent agency (i.e., the Environmental Protection Administration, also abbreviated as EPA) promulgated the regulations governing climate change issues [17]. These regulations include the Air Pollution Control Act, the Climate Change Response Act of 2023 (the former Greenhouse Gas Reduction and Management Act), and the Waste Management Act. The relevant central government agencies also promoted GHG emission reduction. For example, the Ministry of Economic Affairs (MOEA) promulgated a ban on the domestic production of HCFC-22 ($CHClF_2$, one of the refrigerants), thus reducing the production of HFC-23 ($CHF_3$, a by-product in the HCFC-22 manufacturing process) [18]. In addition, the Taiwan government announced "Taiwan's Pathway to Net-Zero Emissions in 2050" on 30 March 2022 [19], which will put emphasis on climate-related legislation as a fundamental base.

Based on the survey of the academic database, few works on the description of the emission trends and regulatory measures of F-gases in Taiwan were discussed in the literature [20–26]. Cheng et al. [26] study $SF_6$ usage and emission factors reflecting common thin-film transistor–liquid crystal display (TFT-LCD) manufacturing practices in Taiwan. Chen and Hu [25] discussed the voluntary F-gas reduction agreement of the semiconductor manufacturing and TFT-LCD industries in Taiwan, showing over 50% reduction rates for the two industries. In the previous studies [20–24], they focused on the environmental risks and policies of HFCs, PFCs, and $NF_3$. In this work, it analyzed the emission trends of F-gases (i.e., HFCs, PFCs, $SF_6$, and $NF_3$) during the period of 2000–2020 by using the updated Taiwan's NIR. In addition, the regulatory strategies and measures will be summarized to be in line with the international protocols on the reduction of the four F-gas emissions. Finally,

the current abatement technologies for F-gas emissions are surveyed by Taiwan's industry alliances like the semiconductor association and the TFT-LCD association.

**Table 1.** Atmospheric lifetime, radiative efficiency, and global warming potential (GWP) of fluorinated greenhouse gases (F-gases) mainly used in Taiwan [1].

| F-Gases [2] | Formula | Atmospheric Lifetime (Year) | Radiative Efficiency (W/(m²-ppb)) | GWP [3] | Main Applications |
|---|---|---|---|---|---|
| HFCs | | | | | |
| HFC-23 | $CHF_3$ | 228 | 0.191 | 14,600 | Etching gas |
| HFC-32 | $CH_2F_2$ | 5.4 | 0.111 | 771 | Refrigerant, Etching gas |
| HFC-41 | $CH_3F$ | 2.8 | 0.025 | 135 | Etching gas |
| HFC-125 | $CHF_2CF_3$ | 30 | 0.234 | 3740 | Refrigerant |
| HFC-134 | $CHF_2CHF_2$ | 10 | 0.194 | 1.260 | Refrigerant |
| HFC-134a | $CH_2FCF_3$ | 14 | 0.167 | 1530 | Refrigerant, aerosol propellant |
| HFC-143 | $CH_2FCHF_2$ | 3.6 | 0.128 | 364 | Refrigerant |
| HFC-143a | $CH_3CF_3$ | 51 | 0.168 | 5810 | Refrigerant |
| HFC-152a | $CH_3CHF_2$ | 1.6 | 0.102 | 164 | Blowing agent, aerosol propellant |
| HFC-227ea | $CF_3CHFCF_3$ | 36 | 0.273 | 3600 | Extinguishing agent |
| HFC-236fa | $CF_3CH_2CF_3$ | 213 | 0.251 | 8690 | Extinguishing agent |
| HFC-245fa | $CHF_2CH_2CF_3$ | 7.9 | 0.245 | 962 | Refrigerant |
| HFC-365mfc | $CH_3CF_2CH_2CF_3$ | 8.9 | 0.228 | 914 | Cleaning solvent, Blowing agent |
| HFC-43-10mee | $CF_3CHFCHFCF_2CF_3$ | 17 | 0.357 | 1600 | Cleaning solvent |
| PFCs | | | | | |
| PFC-14 | $CF_4$ | 50,000 | 0.099 | 7380 | Etching gas |
| PFC-116 | $C_2F_6$ | 10,000 | 0.261 | 12,400 | Etching gas |
| PFC-218 | $C_3F_8$ | 2600 | 0.27 | 9290 | Etching gas |
| PFC-c-318 | $cyc\ (-CF_2CF_2CF_2CF_2-)$ | 3200 | 0.314 | 10,200 | Etching gas |
| PFC-31-10 | $n-C_4F_{10}$ | 2600 | 0.369 | 10,000 | Etching gas |
| PFC-51-14 | $n-C_6F_{14}$ | 3100 | 0.449 | 8620 | Coolant fluid |
| Sulfur hexafluoride | $SF_6$ | 1000 | 0.567 | 24,300 | Insulating gas, Etching gas |
| Nitrogen trifluoride | $NF_3$ | 569 | 0.204 | 17,400 | Etching gas |

[1] The data are from the IPCC report [3]. [2] The selected F-gases are based on the imported statistics [16]. [3] Global warming potential (GWP) for a 100-year time horizon.

## 2. Data Mining Methods

In this study, an analytical description of the emission trends of F-gases (i.e., HFCs, PFCs, $SF_6$, and $NF_3$) from the process industries during the period of 2000–2020 is addressed by using the updated Taiwan's NIR [15], which was announced by the central competent agency (i.e., EPA) in August 2022. Therefore, the authors mainly relied on the qualitative approach by conducting desk research and making trend observations [27]. It should be noted that the data on the Taiwan' NIR were obtained on the basis of the proposed methods in the "2006 IPCC Guidelines for National Greenhouse Gas Inventories" [28]. To calculate the NIR in the sectors and categories, the GHG emission sources are divided into the five sectors in the *2006 IPCC Guidelines*. The industrial process and product use (IPPU) sector is

grouped into the second one. Furthermore, the IPPU sector is categorized as follows: mining industry (2A), chemical industry (2B), metal (aluminum) processing (2C), non-energy products from fuels and solvent use (2D), electronics (manufacturing of semiconductors and optoelectronics) industry (2E), alternatives to ozone-depleting substances (2F), and manufacturing and use of other products (2G). The database can be freely accessed on the official website without permission. To interact with the trend observations of the F-gas emissions, the regulatory strategies and measures for the reduction of the four F-gas emissions are extracted from the relevant regulations, which are established by the Ministry of Justice (MOJ) [29]. They include the Air Pollution Control Act, the Climate Change Response Act, and the Waste Management Act, which were accessed on the official website [29]. Concerning the current abatement technologies for the control of F-gas emissions, they were surveyed by Taiwan's industry alliances [30,31] and the corporate social responsibility (CSR) report of Taiwan's index enterprise [32]. As referred to in the research methods by Borowski [33], the methodological framework of this study is shown in Figure 1.

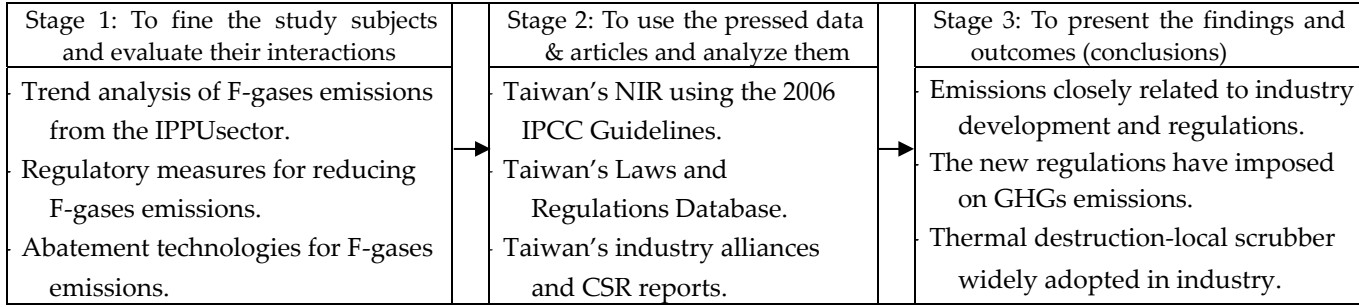

**Figure 1.** Framework items of this study.

### 3. Results and Discussion

*3.1. Analysis of Fluorinated Greenhouse Gases Emissions in Taiwan*

In Taiwan, the majority of F-gas emissions come from the industrial process and product use (IPPU) sector, which includes the chemical industry (2B), metal (aluminum) processing (2C), electronics (manufacturing of semiconductors and optoelectronics) industry (2E), alternatives to ozone-depleting substances (2F), and manufacturing and use of other products (2G). In addition, most HFC substances are used as refrigerants and blowing agents in the energy sector, including in the service, residential, and transport industries. As listed in Table 1, the main applications of these F-gases are refrigerant, cleaning solvent, etching gas, blowing agent, coolant fluid, and extinguishing agent. For the emission trends of the four F-gases, Tables 2 and 3 list the individual emissions and total emissions from the IPPU sector during the period of 2000–2020, respectively [15]. The corresponding percentage variations in F-gas emissions and their total emissions from the IPPU sector every four years are depicted in Figures 2 and 3, respectively [15]. For the total emission of F-gases, it shows a significant increasing trend from 2462 kilotons of carbon dioxide equivalents ($CO_{2eq}$) in 2000 to the peak value (i.e., 12,643 kilotons) of $CO_{2eq}$ in 2004. This trend could be attributed to the rapid development of Taiwan's high-tech industries during the early 2000s. In addition, the regulatory requirements and the industry's voluntary reduction were not strictly imposed on the extensive use of F-gases, causing significant F-gas emissions. Subsequently, it decreases from 10,284 kilotons of $CO_{2eq}$ in 2005 (about 3.54% of the total GHG emissions in 2005) to 3906 kilotons of $CO_{2eq}$ in 2020 (about 1.37% of the total GHG emissions in 2020), down by 69.1% compared to that in 2004. Obviously, the F-gas emission trends are not appropriate to measure the linear correlation with the economic or energy index [34]. The following sections highlight the emission trends of individual F-gases.

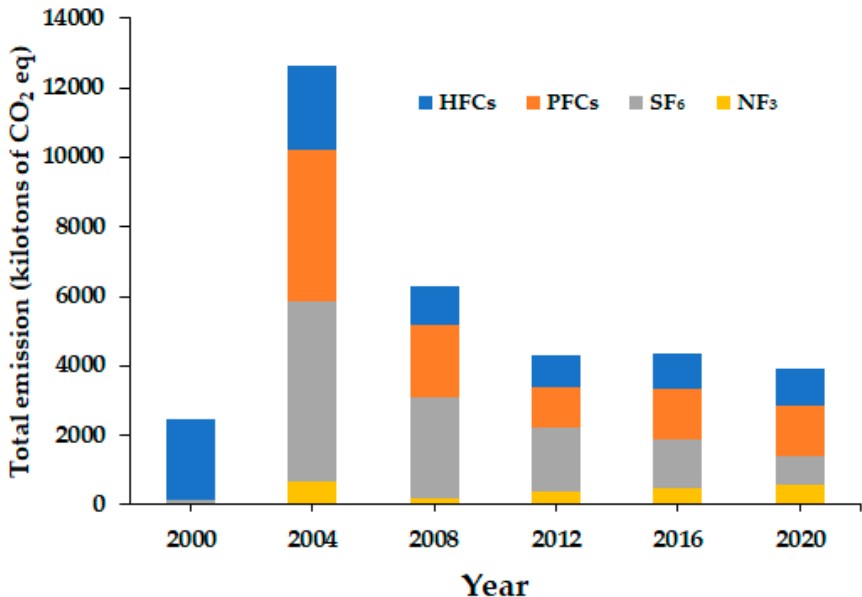

**Figure 2.** Proportional variations in emissions of F-gases in Taiwan during the period of 2000–2020 [15].

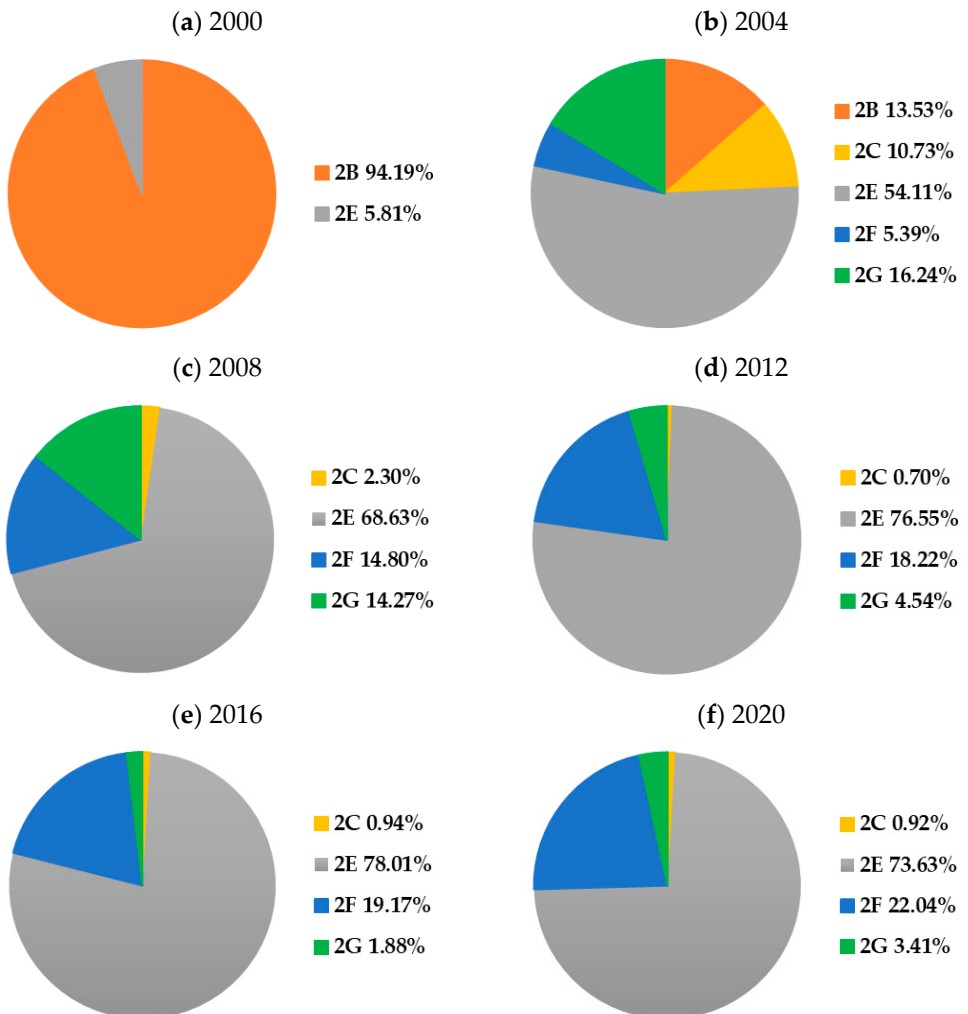

**Figure 3.** Percentages of F-gas emissions from the IPPU sector in Taiwan [15].

**Table 2.** Total emissions of F-gases in Taiwan during the period 2000–2020 [1].

| Year | HFCs | PFCs | $SF_6$ | $NF_3$ | Total |
|------|------|------|--------|--------|-------|
| 2000 | 2319 | 13 | 120 | 10 | 2462 |
| 2001 | 2619 | 2939 | 746 | 235 | 6538 |
| 2002 | 2216 | 4143 | 3914 | 398 | 10,671 |
| 2003 | 2397 | 4198 | 4385 | 540 | 11,520 |
| 2004 | 2451 | 4341 | 5193 | 659 | 12,643 |
| 2005 | 1098 | 3470 | 4951 | 765 | 10,284 |
| 2006 | 1015 | 3664 | 3858 | 688 | 9225 |
| 2007 | 1122 | 3372 | 3381 | 798 | 8673 |
| 2008 | 1074 | 2082 | 2912 | 204 | 6273 |
| 2009 | 1081 | 1560 | 2452 | 577 | 5607 |
| 2010 | 971 | 1770 | 2218 | 258 | 5217 |
| 2011 | 1053 | 1781 | 1918 | 420 | 5172 |
| 2012 | 907 | 1141 | 1852 | 388 | 4288 |
| 2013 | 1019 | 1345 | 1997 | 773 | 5134 |
| 2014 | 1048 | 1556 | 1730 | 667 | 5001 |
| 2015 | 1020 | 1347 | 1523 | 662 | 4552 |
| 2016 | 1026 | 1441 | 1418 | 472 | 4356 |
| 2017 | 1023 | 1409 | 1416 | 440 | 4298 |
| 2018 | 1013 | 1536 | 1302 | 509 | 4360 |
| 2019 | 1027 | 1420 | 935 | 473 | 3855 |
| 2020 | 1053 | 1447 | 842 | 564 | 3906 |

[1] The data from Taiwan's NIR [15]; unit: kilotons of $CO_{2eq}$.

**Table 3.** Total emissions of F-gases from the IPPU sector in Taiwan during the period of 2000–2020 [1].

| Year | Emission Source [2] | | | | | Total |
|------|------|------|------|------|------|-------|
|      | 2B | 2C | 2E | 2F | 2G | |
| 2000 | 2319 | 0 | 143 | 0 | 0 | 2462 |
| 2001 | 2567 | 0 | 3971 | 0 | 0 | 6538 |
| 2002 | 2157 | 1027 | 5544 | 0 | 1943 | 10,671 |
| 2003 | 1937 | 1027 | 6212 | 401 | 1943 | 11,520 |
| 2004 | 1710 | 1357 | 6841 | 682 | 2053 | 12,643 |
| 2005 | 0 | 1063 | 6722 | 996 | 1503 | 10,284 |
| 2006 | 0 | 770 | 6789 | 896 | 770 | 9225 |
| 2007 | 0 | 440 | 6358 | 922 | 953 | 8673 |
| 2008 | 0 | 144 | 4305 | 929 | 895 | 6273 |
| 2009 | 0 | 235 | 3857 | 812 | 703 | 5607 |
| 2010 | 0 | 57 | 4152 | 770 | 238 | 5217 |
| 2011 | 0 | 50 | 3989 | 881 | 252 | 5172 |
| 2012 | 0 | 30 | 3280 | 783 | 195 | 4288 |
| 2013 | 0 | 38 | 4124 | 812 | 160 | 5134 |

**Table 3.** *Cont.*

| Year | Emission Source [2] | | | | | Total |
|------|------|------|------|------|------|-------|
| | 2B | 2C | 2E | 2F | 2G | |
| 2014 | 0 | 33 | 3995 | 828 | 146 | 5001 |
| 2015 | 0 | 43 | 3530 | 851 | 128 | 4552 |
| 2016 | 0 | 41 | 3398 | 835 | 82 | 4356 |
| 2017 | 0 | 59 | 3329 | 821 | 79 | 4298 |
| 2018 | 0 | 81 | 3319 | 811 | 149 | 4360 |
| 2019 | 0 | 43 | 2856 | 846 | 110 | 3855 |
| 2020 | 0 | 36 | 2876 | 861 | 133 | 3906 |

[1] The data from Taiwan's NIR [15]; unit: kilotons of $CO_{2eq}$. [2] Emission source notations: 2B—chemical industry; 2C—metal process; 2E—electronics industry; 2F—alternatives to ozone-depleting substances; 2G—manufacturing and use of other products.

### 3.1.1. Hydrofluorocarbons (HFCs)

In general, HFCs containing one carbon atom (i.e., HFC-23, HFC-32, and HFC-41) are mostly used as etching gases in the semiconductor manufacturing and TFT-LCD industries. In contrast, HFCs containing two carbon atoms (i.e., HFC-125, HFC-134, HFC-134a, HFC-143, and HFC-143a) are often used as refrigerants in the air-conditioning appliances of residences and vehicles. As indicated in Table 1, HFC-32 is also used as a refrigerant because of its low GWP (i.e., 771). For example, the commercial refrigerant, R410A, is a mixture of HFC-125 (50%), and HFC-32 (50%). Another commercial refrigerant R407C is a mixture of HFC-125 (25%), HFC-134a (52%), and HFC-32 (23%). Obviously, Table 2 shows the two different stages of HCF emissions in Taiwan. During the period 2000–2004, the total HFC emissions approximately ranged from 2200 to 2600 kilotons of $CO_{2eq}$. However, the total HFCs emissions are approximate to 1000 kilotons of $CO_{2eq}$ since 2005. In accordance with the Montreal Protocol, the Taiwan government has promulgated a ban on the production of refrigerant HCFC-22 since 2005, which was only produced by a chemical company. During the manufacture of HCFC-22, a by-product, HFC-23, will be generated and emitted from the process. Its emission is obtained by multiplying the HCFC-22 production amount with its default emission factor (i.e., 1.4%). As seen in Table 3, the emission amounts of F-gases from the 2B source have been null since 2005. Since then, the total F-gas (i.e., HFC) emission amounts have maintained a stable trend because the emission source is mainly derived from the 2F industry. The emission amounts of HFCs refrigerants were estimated by leakage rates proposed by the IPCC method [28]. Regarding the emission amounts of HFCs for other uses (i.e., blowing agent, cleaning solvent, and extinguishing agent), they were estimated by their imported amounts but only accounted for less amounts. Concerning the data on the percentages of HFCs (Figure 3), they were 94.2% in 2000, 19.4% in 2004, 17.1% in 2008, 21.2% in 2012, 23.6% in 2016, and 27.0% in 2020. The significant decrease during the early 2000s could be due to the restricted use of HFCs.

### 3.1.2. Perfluorocarbons (PFCs)

PFCs are known for their rather stable properties because of the strength of the carbon-fluorine bond, leading to their industrial applications in the cleaning of dry etch and chemical vapor deposition (CVD) processes. Therefore, the main emission sources of PFC are from the semiconductor manufacturing and TFT-LCD industries in Taiwan. As listed in Table 1, these PFCs substances, including PFC-14 ($CF_4$), PFC-116 ($C_2F_6$), PFC-218 ($C_3F_8$), PFC-c-318 ($C_4F_8$, octafluorocyclobutane), and PFC-51-14 ($C_6F_{14}$, perfluorohexane), are potent GHGs due to their long atmospheric lifetimes and high GWP values. The total emissions of PFCs in Taiwan indicate a decreasing trend over the past two decades. Table 2 shows a decline rate of about 75% (i.e., 4341 kilotons of $CO_{2eq}$ in 2004 vs. 1447 kilotons of $CO_{2eq}$ in 2020). It can also be seen that the total emissions of PFCs increased significantly

from 13 kilotons of $CO_{2eq}$ in 2000 to 4198 kilotons of $CO_{2eq}$ in 2003. These variations are attributed to the mass production of Taiwan's electronic industries during the early 2000s, which subsequently took actions on voluntary PFC reduction technologies like de-PFCs/local scrubbers, and alternatives with low GWP. Also shown in Figure 2, the proportions of PFC emissions account for about one-third of the total F-gas emissions since 2004.

### 3.1.3. Sulfur Hexafluoride ($SF_6$)

The main emission sources of $SF_6$ include the electrical power, semiconductor manufacturing, TFT-LCD, and magnesium production industries. Due to its high inertness and unique dielectric properties, it has been used as a dielectric medium in electric power systems, a silicon etchant for wafer manufacturing, and an inert gas for the casting of magnesium. Based on the data in Table 2 and Figure 2, the trends of total $SF_6$ emissions are very similar to those of total PFC emissions because voluntary emission reduction actions have also been taken by the industrial sector. It shows a decline rate of over 80% since 2004 (i.e., 5193 kilotons of $CO_{2eq}$ in 2004 vs. 842 kilotons of $CO_{2eq}$ in 2020). Herein, the emission source of $SF_6$ from magnesium production in Taiwan can be negligible due to the limited domestic production since the mid-2000s. As seen in Table 3, the emission amounts of the F-gases from the chemical industry (2C) indicate a gradual decline since 2006 because of magnesium production moving out and $SF_6$ reduction by process change. In addition, the total $SF_6$ emission amounts also show a decreasing trend due to the recovery/recycling technologies widely adopted by the electrical power industry (2G). Using the data on the total F-gas emissions, the proportions of $SF_6$ emissions (Figure 3) also show a significant increase from 4.9% in 2000 to 46.4% in 2008, but they decreased subsequently to 21.5% in 2020.

### 3.1.4. Nitrogen Trifluoride ($NF_3$)

Nitrogen trifluoride ($NF_3$) is primarily used to remove silicon particles and silicon-containing compounds during the manufacturing of semiconductor devices like flat-panel displays, photovoltaics, and light-emitting diodes (LEDs). Although its atmospheric lifetime (i.e., 569 years) is much smaller than other PFCs (2600–50,000 years) and $SF_6$ (1000 years), it is a potent GHG with a high GWP (i.e., 17,400) and mass consumption in the semiconductor manufacturing and TFT-LCD industries since the early 2000s, thus grouping with effect from 2013 and the commencement of the second commitment period of the Kyoto Protocol [8,35]. Just like the increasing trends of PFCs and $SF_6$ emissions, the total emissions of $NF_3$ increased significantly from 10 kilotons of $CO_{2eq}$ in 2000 to 798 kilotons of $CO_{2eq}$ in 2007. Thereafter, the total emissions of $NF_3$ indicate a jagged pattern, ranging from 200 to 800 kilotons of $CO_{2eq}$ during the period of 2007–2020. Herein, the sharp decline of the total $NF_3$ emission (i.e., 204 kilotons of $CO_{2eq}$) in 2008 should be attributed to the 2008 economic recession around the world. On the other hand, the ratios of total $NF_3$ emission amounts to total F-gas emission amounts (Figure 2) indicate a rising trend due to the reduction in $SF_6$ use; that is, 0.4% in 2000, 3.2% in 2004, 3.2% in 2008, 9.0% in 2012, 10.8% in 2016, and 14.4% in 2020.

### 3.2. Regulatory Strategies for Controlling the Emissions of Fluorinated Greenhouse Gases

In Taiwan, the regulatory strategies for the control of fluorinated GHG (F-gas) emissions are authorized by the Air Pollution Control Act, the Climate Change Response Act, and the Waste Management Act. The relevant measures will be further addressed in the following sub-sections.

### 3.2.1. Air Pollution Control Act

This Act was recently revised on 1 August 2018. Under the authorizations of the Act, the Taiwan EPA promulgated the six GHGs as air pollutants on 9 May 2012, including $CO_2$,

$CH_4$, $N_2O$, HFCs, PFCs, and $SF_6$. Article 20 of the Act refers to the countermeasures against these air pollutants [17], including:

- The stationary sources (e.g., vents or pipelines) in the process industries shall comply with the emission standards of designated air pollutants by installing a closed vent/collection system and an air pollution control system;
- According to the specially designated industry categories, facilities, pollutant items, or areas, the EPA, in consultation with relevant agencies (i.e., the Ministry of Economic Affairs), shall determine the emission standards.

For example, the EPA promulgated the emission standards for volatile organic compounds (VOCs) in the regulations ("Air Pollution Control and Emission Standards for Semiconductor Manufacturing Industry" and "Air Pollution Control and Emission Standards for Optoelectronic Materials and Element Manufacturing Industry"). Herein, the so-called VOCs include HFCs and PFCs, but not carbon dioxide ($CO_2$) or methane ($CH_4$). With the promulgation of the Greenhouse Gas Reduction and Management Act (GGRMA) of 2015, the provisions for controlling GHGs in the Air Pollution Control Act will comply with the CCRA.

### 3.2.2. Climate Change Response Act

To achieve carbon neutrality by 2050, the Taiwan government replaced the Greenhouse Gas Reduction and Management Act (GGRMA) of 2015 with the Climate Change Response Act (CCRA) passed on 15 February 2023 [29]. The new Act defines greenhouse gases (GHGs), which refer to the following substances: $CO_2$, $CH_4$, $N_2O$, HFCs, PFCs, $SF_6$, $NF_3$, and other GHGs designated by the central competent authority (i.e., EPA). The most noteworthy point is to set a long-term national net-zero emission (carbon neutrality) policy by 2050. To reach the goal, all levels of the Taiwan government shall implement GHG reduction strategies and also develop negative emission technologies. As mentioned in Section 3.1, the major emission sources of the four F-gases were from the industrial (or manufacturing) and waste management sectors in Taiwan. With the joint efforts of the central competent authorities like the Ministry of Economic Affairs (MOEA) and EPA, the key strategies will focus on the following orientations:

- Process modification: Equipment renewal by phase-out, fluorinated gases (F-gases) reduction by environment-friendly substitutes, and recovery/recycling/destruction systems installed in the industrial sector;
- Circular economy: Recovery/recycling/storage systems installed in the waste (waste electronic appliances like air conditioners and refrigerators, and waste vehicles) management sector.

### 3.2.3. Waste Management Act

As mentioned above, F-gases (especially HFCs) have been widely used as refrigerants in vehicles and air conditioners. In addition, some HFCs are used as blowing agents in refrigerators. Therefore, these articles will emit these F-gases while being discarded as waste electrical and electronic equipment (WEEE) and scrap cars [14,36]. In Taiwan, the legal system for the recycling and treatment of waste is based on the Waste Management Act, which aims to define the categories of waste and clarify its obligations and responsibilities. Herein, waste refers to any movable solid or liquid substance or object due to some discarded reasons like weakened performance and no economic or market value. In response to extended producer responsibility (EPR) and sustainable material management, the Taiwan government has been implementing zero waste and resource recycling promotion programs since 1997 [37,38]. Based on the authorization of the Act, the regulated recyclable wastes are defined as those that could cause serious environmental pollution and also have value for recycling and reuse. In this regard, the Taiwan EPA has announced the regulated recyclables, including plastic containers, metal (iron/aluminum) containers, vehicles, tires, batteries, home electric appliances, information technology (IT) products, and lighting. Concerning the recovery of refrigerant and blowing agent from waste home

electric appliances, the Taiwan EPA promulgated the regulations ("Facilities Standards for the Recycling Storage, and Disposal of Electrical and Electronic Waste" and "Facilities Standards for the Recycling Storage, and Disposal of Waste Vehicles"). These regulations require the recycling enterprises to install the F-gas recovery (liquefaction or adsorption) equipment and storage tank for refrigerants in the air conditioning system and blowing agents in the refrigerator foam insulation system [39,40].

### 3.3. Survey of Current Abatement Technologies for Controlling the Emissions of Fluorinated GHGs

Concerning the abatement technologies for the control of fluorinated GHG emissions, there are many different options for mitigating their emissions from the process industries, especially in the semiconductor manufacturing and TFT-LCD industries. They are basically grouped into three different approaches [20,41]: (1) material modification and/or substitution; (2) capture recovery-recycling technology; and (3) thermal destruction-local scrubbing technology. The first method should be prioritized, but it could be limited to using more environmentally friendly refrigerants and cleaning solvents. With the adoption of the Significant New Alternatives Policy (SNAP) database set by the US Environmental Protection Agency [42], the available environment-friendly refrigerants with low GWP and high safety include hydrofluoro-olefins (HFOs) [43,44], hydrochlorofluoro-olefins (HCFOs), and hydrofluoro-ethers (HFEs). According to the database of SNAP [42], the newly commercial refrigerants include HFO-1234yf (2,3,3,3-tetrafluoroprop-1-ene), HCFO-1233zd(E) (trans-1-chloro-3,3,3-trifluoroprop-1- ene), and HFE-347mcc3 (heptafluoropropyl methyl ether). It should be noted that these acceptable refrigerants (e.g., R-448A) may be a marketed mixture of HFCs and HFOs. Regarding the environment-friendly cleaning solvents, the fluorinated substances, including HFOs, HCFOs, and HFEs, are listed in the SNAP program. These acceptable substitutes, including HFO-1336mzz(Z) ((Z)-1,1,1,4,4,4-hexafluorobut-2-ene), HCFO-1233zd(E), and HFE-347mcc3, have been specified and also used in industrial processes. The second method is to apply capture, recovery, and recycling technologies, including cryogenic condensation, adsorption, and membrane separation. However, their recovery availabilities are less common in industrial processes because of the physico-chemical properties of target F-gases (i.e., high fugacity, high stability, low polarity, and low solubility in water), the complicated composition of vent gas, and the high purification requirements of recovered F-gases. Therefore, the most commonly used method for the reduction of F-gas emissions from the process is to adopt the thermal destruction-local scrubbing technology. These de-PFC/local scrubber systems have been installed in Taiwan's semiconductor manufacturing process [32]. Currently, the thermal destruction methods include the following types: plasma-based, catalyst-based, or combustion-based. Taking $NF_3$ destruction as an example [23], the following equations show its possible stoichiometry reactions with oxygen, hydrogen, and moisture (water vapor).

$$2NF_3 + 2O_2 \rightarrow NO + 3F_2$$

$$2NF_3 + 3H_2 \rightarrow N_2 + 6HF$$

$$2NF_3 + 3H_2O \rightarrow 6HF + NO + NO_2$$

These thermal decomposition products, or by-products, are easily dissolved in water. For this reason, the vent exhausts from the thermal destruction unit are further removed by the wet scrubbing unit. Although the decomposition product NO is only slightly soluble in water, it is apt to form $NO_2$ in an oxidative environment. Obviously, these decomposition products are acidic compounds, which will react with an alkaline solution to produce fluoride like NaF. For example, sodium hydroxide (NaOH), a strong base, will react with HF, $F_2$, or $NO_2$ molecules to form the non-toxic substances, which can be illustrated below.

$$HF + NaOH \rightarrow NaF + H_2O$$

$$2F_2 + 4NaOH \rightarrow 4NaF + O_2 + 2H_2O$$

$$2NO_2 + 2NaOH \rightarrow NaNO_2 + NaNO_3 + H_2O$$

Furthermore, the by-product fluoride or fluoride-containing wastewater in semiconductor or optoelectronic industries may be converted to a valuable product (cryolite, $Na_3AlF_6$) by a crystallization process [45].

## 4. Conclusions

Fluorinated greenhouse gases (F-gases) have been widely used as industrial and commercial products, such as refrigerant, blowing agent, cleaning solvent, etching gas, coolant fluid, and extinguishing agent. This desk research presents the trends of F-gases (i.e., HFCs, PFCs, $SF_6$, and $NF_3$) emissions and their sources from the industrial process and product use (IPPU) sector over the past two decades (2000–2020) based on Taiwan's GHG inventory report. The findings show a significant increasing trend from 2462 kilotons of carbon dioxide equivalents ($CO_{2eq}$) in 2000 to the peak value (i.e., 12,643 kilotons) of $CO_{2eq}$ in 2004, which is consistent with the rapid development in Taiwan's semiconductor manufacturing and TFT-LCD. Afterwards, it decreases from 10,284 kilotons of $CO_{2eq}$ in 2005 (about 3.54% of the total GHG emissions in 2005) to 3906 kilotons of $CO_{2eq}$ in 2020 (about 1.37% of the total GHG emissions in 2020), down by 69.1% compared to that in 2004. Obviously, an important conclusion from the preliminary time-series analysis is that the ongoing efforts towards the regulatory requirements and the industry's voluntary reduction strategies cause a significant reduction of F-gas emissions in Taiwan. Based on the survey of the abatement technologies for controlling the F-gas emissions in Taiwan's high-tech industries, the current options focused on thermal destruction local scrubbing systems. With the enforcement of the Kigali Amendment for adding HFCs to the list of phase-outs since 1 January 2019, environmentally friendly refrigerants with low GWP and high safety, including hydrofluoro-olefins (HFOs), hydrochlorofluoro-olefins (HCFOs), and hydrofluoro-ethers (HFEs), will be more and more used in refrigeration and air conditioning in the near future. Another notable conclusion is that the thermal destruction-local scrubber approach will be the mainstream technology for the reduction of PFCs, $SF_6$, and $NF_3$ emissions. However, these de-PFC/local scrubber systems must pay special attention to the efficient removal of toxic by-products from the vent gas.

**Author Contributions:** Conceptualization, W.-T.T.; data collection, C.-H.T.; data analysis, C.-H.T.; writing—original draft preparation, W.-T.T.; writing—review and editing, W.-T.T. All authors have read and agreed to the published version of the manuscript.

**Funding:** This research received no external funding.

**Institutional Review Board Statement:** Not applicable.

**Informed Consent Statement:** Not applicable.

**Data Availability Statement:** The authors confirm that the data supporting the findings of this study are available within the article.

**Conflicts of Interest:** The authors declare no conflict of interest.

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
