# Peer review of "A Survey on Fluorinated Greenhouse Gases in Taiwan: Emission Trends, Regulatory Strategies, and Abatement Technologies"

_environments, doi:10.3390/environments10070113_

Round 1
Reviewer 1 Report
The paper provides an nice overview on the status of emissions of fluorinated GHGs in Taiwan based on the national inventory report, and provides information on the regulatory regimes and technological options, which may reduce emission of these harmful gases. In my opinion, the paper does not add to existing knowledge, whereby I cannot recommend publication.
Language is ok, but use of terms such as "excellent physiochemical properties" (excellent as to what?) and "roaring emissions" (emissions don´t roar) should be avoided.
Author Response
Q1. Language is ok, but use of terms such as "excellent physiochemical properties" (excellent as to what?) and "roaring emissions" (emissions don´t roar) should be avoided.
Reply: As suggested by the reviewer, some terms have been checked and revised to make them proper.
Reviewer 2 Report
This is a good article for introducing fluorinated fluids.
I would like to remind the authors that PFCs are also widely used as coolant fluids for semiconductor fabrication tools.
Considering the more and more strict regulations against these chemicals, any candidates the authors can recommend to substitute them?
Thanks!
Author Response
Q1. I would like to remind the authors that PFCs are also widely used as coolant fluids for semiconductor fabrication tools.
Reply: As suggested by the reviewer, the application of PFCs as coolant fluids for semiconductor fabrication tools has been added to the manuscript, including Table 1.
Q2. Considering the more and more strict regulations against these chemicals, any candidates the authors can recommend to substitute them?
Reply: As suggested by the reviewer, the candidates for PFCs substitutes have been addressed to meet stricter regulations against these chemicals. According to the database of Significant New Alternatives Policy (SNAP), the new-generation commercial refrigerants include HFO-1336mzz(Z) ((Z)-1,1,1,4,4,4-hexafluorobut-2-ene), HCFO-1233zd(E) (trans-1-chloro-3,3,3- trifluoroprop-1-ene) and HFE-347mcc3 (heptafluoropropyl methyl ether). These PFCs substitutes have been added to the manuscript (Sec. 3.3).
Reviewer 3 Report
Highlight changes in yellow in a next revision, please. No track changes.
Dear authors, unless similarity is addressed, there is no proper review. I do not refer to the preprints.
And when the preprints also contain similarity, that is a major issue.
General comments then:
Move away from similarity, serious issue
Use your own wording
do not start abstract with “due”
needs to rely on recent data…
“tons of CO2eq in 2020.”
why past?
“Table 1 summarized”
This is no proper refencing style:
“1 Source [3]. 2 Global warming potential (GWP) for 100-year time horizon. 3 Based on the imported statistics in recent years [13].”
Methods says very little, also with similarity.
Needs huge extension…
There can be no similarity in results…
data would need to include recent years…
again no proper referencing…
“Table 2. Total emissions of fluorinated greenhouse gases since 2000 in Taiwan. 1”
until when? be clear in captions:
“Figure 1. Proportion variations on emissions of fluorinated greenhouse gases (F-gases) in Taiwan since 2000.”
similarity all over, not possible:
“3.2. Regulatory strategies for controlling the emissions of fluorinated greenhouse gases”
Even in the conclusions.
I believe the authors need to understand that there is no merit in similarity. The article needs to be made relevant.
It is not, as it is now, in my opinion, nor is the data updated…
“the period of 2000-2020 have been analyzed.”
This makes no sense to me, in terms of “ending” the article.
Conclusions need to translate methodology, clear findings, practical implications, limitations and future prospects.
the volume of references is low, references from relevant authors would need to be there, from 2023 and more international too
Moderate
Author Response
Q1. Move away from similarity, serious issue. Use your own wording
Reply: As suggested by the reviewer, the manuscript has been revised to reduce its similarity with the previous publications.
Q2. do not start abstract with “due”, needs to rely on recent data…“tons of CO2eq in 2020.”
Reply: As suggested by the reviewer, the abstract has been revised to use the proper words and/or phrases.
Q3. why past? “Table 1 summarized”
Reply: As suggested by the reviewer, the present tense is used in the sentence.
Q4. This is no proper refencing style:
“1 Source [3]. 2 Global warming potential (GWP) for 100-year time horizon. 3 Based on the imported statistics in recent years [13].”
Reply: As suggested by the reviewer, the reference style has been revised to use the proper notation.
Q5. Methods says very little, also with similarity. Needs huge extension…There can be no similarity in results
Reply: As suggested by the reviewer, the section (Methods) has been extended by drawing the methodological framework of this study (Figure 1) in the revised manuscript.
Q6. data would need to include recent years…again no proper referencing… “Table 2. Total emissions of fluorinated greenhouse gases since 2000 in Taiwan. 1”
Reply: As suggested by the reviewer, the title in Table 2 has been revised to make it proper. According to the updated NIR in Taiwan, the data on F-gas emissions presented only up to 2020.
Q7. until when? be clear in captions: “Figure 1. Proportion variations on emissions of fluorinated greenhouse gases (F-gases) in Taiwan since 2000.”
Reply: As suggested by the reviewer, the caption in Figure 1 (changed to Figure 2) has been revised to make it proper.
Q8. Similarity all over, not possible: “3.2. Regulatory strategies for controlling the emissions of fluorinated greenhouse gases”
Reply: As suggested by the reviewer, the sec. 3.2 has been revised to reduce its similarity with the previous publications.
Q9. Even in the conclusions. I believe the authors need to understand that there is no merit in similarity. The article needs to be made relevant. It is not, as it is now, in my opinion, nor is the data updated…“the period of 2000-2020 have been analyzed.” This makes no sense to me, in terms of “ending” the article. Conclusions need to translate methodology, clear findings, practical implications, limitations and future prospects.
Reply: As suggested by the reviewer, the Conclusion has been extensively revised to cover methodology, clear findings, practical implications, limitations and future prospects.
Q10. the volume of references is low, references from relevant authors would need to be there, from 2023 and more international too
Reply: As suggested by the reviewer, the relevant and updated references (Refs. 4, 5, 6, 33, 40, 41) have been added to the manuscript.

Reviewer 4 Report
Dear Authors,
Thank you for possibility to read your interesting paper titled: A survey on fluorinated greenhouse gases in Taiwan: Emission trends, regulatory strategies, and abatement technologies. After reading your paper I have some comments, suggestions and advice.
Please write section methods; which methods did you use to make your research. The most popular meaning of survey is: a data collection tool that lists a set of structured questions to which respondents provide answers based on their knowledge and experiences. It is a standard data gathering process that allows you to access information from a predefined group of respondents during research. In your case it is better to use desk-research method (secondary analysis of data). Please read some papers where methods were described e.g. Mitigating Climate Change and the Development of Green Energy versus a Return to Fossil Fuels Due to the Energy Crisis in 2022. Energies, 15(24), 9289 or Research methods. Journal of Business & Economics Research (JBER), 5(3).
Authors wrote: Based on the survey by the database like Web of Science. It is better to use the word study, database investigation, this is desk-research method of the study.
Table 3- I don’t understand, why Authors used 2B, 2C etc for the different types of industry? Moreover, please write what is interesting from that results? What is the contribution of that Figures. For example: Why in 2004 there is pick of GHG emission (Fig.1).
Fig.2 – You don’t need to repeat in the title the years (a) 2000 etc, because you already put years on the Figure. Why there are differences in the structure between 2012 and 2016? Why in 2012 there is huge value of electronics industry? Please try to show some interesting and valuable description of the situation. Please read some papers which show, how the variables can be interpretated e.g. Water and Hydropower—Challenges for the Economy and Enterprises in Times of Climate Change in Africa and Europe. Water, 14(22), 3631 or Energy of feeding and chopping of biomass processing in the working units of forage harvester and energy balance of methane production from selected energy plants species. Biomass and Bioenergy, 128, 105301.
Please show what is the contribution of your research to the development of this issue. Are there any relationships/correlations between the variables? Please show your own analysis and thoughts. Currently, the facts from the literature are shown, but what are the results?
Please read more scientific papers concerning the topic from years 2022-2023.
Author Response
Q1. Please write section methods; which methods did you use to make your research. The most popular meaning of survey is: a data collection tool that lists a set of structured questions to which respondents provide answers based on their knowledge and experiences. It is a standard data gathering process that allows you to access information from a predefined group of respondents during research. In your case it is better to use desk-research method (secondary analysis of data). Please read some papers where methods were described e.g. Mitigating Climate Change and the Development of Green Energy versus a Return to Fossil Fuels Due to the Energy Crisis in 2022. Energies, 15(24), 9289 or Research methods. Journal of Business & Economics Research (JBER), 5(3).
Reply: As suggested by the reviewer, the section (Methods) has been extended by drawing the methodological framework of this study (Figure 1) in the revised manuscript.
Q2. Authors wrote: Based on the survey by the database like Web of Science. It is better to use the word study, database investigation, this is desk-research method of the study.
Reply: As mentioned above, the section (Methods) has been revised to make it proper and clear in the revised manuscript.
Q3. Table 3- I don’t understand, why Authors used 2B, 2C etc for the different types of industry? Moreover, please write what is interesting from that results? What is the contribution of that Figure. For example: Why in 2004 there is pick of GHG emission (Fig.1).
Reply: In Taiwan, the national greenhouse gas inventories were based on the proposed methods of the “2006 Intergovernmental Panel on Climate Change Guidelines (2006 IPCC Guidelines)”. Therefore, the emission source notations (2B, 2C etc.) in the IPPU sector were adopted in this work. In addition, some comments derived from Table 3 and Figure 1 were also interpreted to make them valuable and clear.
“……. This trend could be attributed to the rapid development in the Taiwan’s high-tech industries during the early 2000. In addition, the regulatory requirements and the industry’s voluntary reduction strategies were not strictly imposed on the extensive use of F-gases, also causing the significant F-gases emissions. ……”
Q4. Fig.2 – You don’t need to repeat in the title the years (a) 2000 etc, because you already put years on the Figure. Why there are differences in the structure between 2012 and 2016? Why in 2012 there is huge value of electronics industry? Please try to show some interesting and valuable description of the situation. Please read some papers which show, how the variables can be interpreted e.g. Water and Hydropower—Challenges for the Economy and Enterprises in Times of Climate Change in Africa and Europe. Water, 14(22), 3631 or Energy of feeding and chopping of biomass processing in the working units of forage harvester and energy balance of methane production from selected energy plants species. Biomass and Bioenergy, 128, 105301
Reply: As suggested by the reviewer, the caption in Figure 2 (changed to Figure 3) has been revised to make it proper. In addition, some comments derived from Figure 3 were also interpreted to make them valuable and reasonable. Regarding the differences between 2012 and 2016 in the electronics industry (2E), the colours in the pie charts have been corrected to make them consistent. To avoid the misunderstanding, the same industry adopted the same colour in 2000, 2004, 2008, 2012, 2016, and 2020.
Q5. Please show what is the contribution of your research to the development of this issue. Are there any relationships/correlations between the variables? Please show your own analysis and thoughts. Currently, the facts from the literature are shown, but what are the results?
Reply: As suggested by the reviewer, the description about the contribution of this research to the development of this issue has been added to the Conclusions.
“……. This work may be the first study on the trends of F-gases (i.e., HFCs, PFCs, SF6 and NF3) emissions and their sources from the industrial process and product use (IPPU) sector over the past two decades (2000-2020) based on the Taiwan’s GHG inventory report. The findings showed significant increasing trend from 2,462 kilotons of carbon dioxide equivalents (CO2eq) in 2000 to the peak value (i.e., 12,643 kilotons) of CO2eq in 2004 due to the rapid development in the Taiwan’s semiconductor manufacturing and TFT-LCD. Subsequently, it decreased from 10,284 kilotons of CO2eq in 2005 (about 3.54% of the total GHG emissions in 2005) to 3,906 kilotons of CO2eq in 2020 (about 1.37% of the total GHG emissions in 2020), down by 69.1% compared to that in 2004. Obviously, these achievements were closely related to the ongoing efforts towards the regulatory requirements and the industry’s voluntary reduction strategies, causing the significant reduction of F-gases emissions. ……”
Q6. Please read more scientific papers concerning the topic from years 2022-2023.
Reply: As suggested by the reviewer, the relevant and updated references (Refs. 4, 5, 6, 33, 41) have been added to the manuscript.

Round 2
Reviewer 1 Report
I commend the authors for the improvements made
Author Response
Point 1: I commend the authors for the improvements made.
Response 1: Thanks for the positive feedback.
Reviewer 3 Report
Highlight changes in yellow in a next revision, please. No track changes.
Not enough.
“Q1. Move away from similarity, serious issue. Use your own wording
Reply: As suggested by the reviewer, the manuscript has been revised to reduce its similarity with the previous publications. ”
See that significant similarity is present in several parts of the manuscript without any reference in many cases.
I expect authors to understand the importance of addressing this otherwise the paper is compromised.
Also present in the conclusions, as mentioned before, although revised.
This is a major issue…
Again then:
Move away from similarity…
It needs to be dropped!
Use your own wording…
I would ask the authors to check the manuscript again. What are environmental properties…?
“Table 1 summarizes the main environmental properties (i.e., atmospheric lifetime, radiative efficiency and global warming potential)”
Why this term?!
“Table 1 summarizes the main environmental properties (i.e., atmospheric lifetime, radiative efficiency and global warming potential)”
Moderate
Author Response
Point 1: Significant similarity is present in several parts of the manuscript without any reference in many cases.
Response 1: Based on the survey results by the Turnitin system, the manuscript has been again revised to reduce its similarity by citing the proper references and also using the own wording.
Point 2: Also present in the conclusions, as mentioned before, although revised.
Response 2: As suggested by the reviewer, the Conclusion has been again revised to meet the writing requirement.
Point 3: What are environmental properties…?
“Table 1 summarizes the main environmental properties (i.e., atmospheric lifetime, radiative efficiency and global warming potential)”
Response 3: To present the data relevant to the greenhouse effect, the term “environmental properties” has been changed to “Atmospheric lifetime, radiative efficiency and global warming potential (GWP)” in the revised manuscript.

Reviewer 4 Report
Dear Authors,
You made small changes (cosmetics changes) but still there is no changes which will increase the value of that paper. I asked you to read what methodology means, what kind of methods and technics you can use. In your Figure you wrote in column Methods: inventory reports, regulations database... these elements are not methods, this are parts (sources) on deask research methods. Please read some more papers (I indicated you one good paper, where you can see the methods description and you can read more papers with that informations/description where it was mentioned/written about methods). I suggested that you read a few papers and use the knowledge from these articles and support your paper with the papers you read. When you use some papers, please make citations, that you supported your paper by these articles. Other readers can learn from indicated articles.
Regarding 2B, 2C...still don't know why you use these abbreviations? Why 2B, 2C and not 3B, 3C? Where are these from? Why B,C and not J,K?
I asked to do some statistical analysis to show that the article is not just a collection of descriptions taken from sources, but something that brings something new to science. What is your scientific contribution? I also provided an article that you should read and see how it should be done. Please carefully and conscientiously read my previous review again and search the articles I have provided and look for more where you will find how to present your results.
There is still work to be done for deep improvement. I hope the next version will be correct. Please make really better version. Good luck
Author Response
Point 1: I asked you to read what methodology means, what kind of methods and technics you can use. In your Figure you wrote in column Methods: inventory reports, regulations database... these elements are not methods, this are parts (sources) on desk research methods. Please read some more papers (I indicated you one good paper, where you can see the methods description and you can read more papers with that information/description where it was mentioned/written about methods). I suggested that you read a few papers and use the knowledge from these articles and support your paper with the papers you read. When you use some papers, please make citations, that you supported your paper by these articles. Other readers can learn from indicated articles.
Response 1: The authors agree to the reviewer’s suggestion. The desk research is a method of exploring data from the pressed (or open-accessed) documents and articles relevant to a particular topic. As provided by the reviewer, we also cited the two papers to support my paper in the section of “Data Mining Methods”. More importantly, the methodological scheme of this study is redrawn and shown in Figure 1.
Point 2: Regarding 2B, 2C...still don't know why you use these abbreviations? Why 2B, 2C and not 3B, 3C? Where are these from? Why B,C and not J,K?
Response 2: In this work, the authors analyzed the data on the Taiwan’s National Inventory Report (NIR), which was based on the “2006 IPCC Guidelines for National Greenhouse Gas Inventories”. To calculate the NIR in the sectors and categories, the GHG emission sources are divided into the five sectors in the 2006 IPCC Guidelines. The industrial process and product use (IPPU) sector is grouped into the second one. Furthermore, the IPPU sector is categorized as follows: mining industry (2A), chemical industry (2B), metal (aluminum) process (2C), non-energy products from fuels and solvent use (2D), electronics (manufacturing of semiconductor and optoelectronics) industry (2E), alternatives to ozone-depleting substances (2F), manufacturing and use of other products (2G).
Point 3: I asked to do some statistical analysis to show that the article is not just a collection of descriptions taken from sources, but something that brings something new to science. What is your scientific contribution? I also provided an article that you should read and see how it should be done. Please carefully and conscientiously read my previous review again and search the articles I have provided and look for more where you will find how to present your results.
Response 3: As suggested by the reviewer, the provided paper has been cited in the Sec. 3.1 of the revised manuscript. The description about the contribution of this desk research to the development of this topic has been added to the Conclusions. Regarding the statistical analysis, this work only analyzed the F-gases emission trends during the period of 2000-2020 based on the categories of the IPPU sector, not proper to measure the linear correlation with the economic index (e.g., GDP) or energy index (e.g., energy efficiency, or energy efficiency per capita). However, the significant findings are addressed in the Sec. 3.1, showing that the F-gases emissions in time sequence are closely related to the industrial development and the regulatory requirements.

Round 3
Reviewer 3 Report
Highlight changes in yellow in a next revision, please. No track changes.
The similarity is still present. I will not check the manuscript again…
In such cases, the subcaptions must be present after the main caption, by letter…
“Figure 3. Percentages of F-gases emissions from the IPPU sector in Taiwan [15].”
Moderate
Reviewer 4 Report
Dear Authors, you took under consideration my comments and sugestiom. Now the paper much better and can be published.